# Characteristics of Mother–Daughter Relationships and Sexual Risk-Coping Consciousness among Japanese Female University Students

**DOI:** 10.3390/ijerph17238795

**Published:** 2020-11-26

**Authors:** Chisato Yamanaka, Kimiko Kawata

**Affiliations:** 1Fukuoka Birth Clinic, Fukuoka 819-0006, Japan; mc0420m@gmail.com; 2Department of Health Sciences, Division of Nursing Sciences, Faculty of Medical Sciences, Kyushu University, Fukuoka 812-8582, Japan

**Keywords:** mother-daughter relationship, sexual risk-coping consciousness, female university students

## Abstract

For heterosexual Japanese women in their early 20s, it is important to maintain good sexual health, develop intimate relationships with a partner, and go through the process of having a family. This study aimed to determine the characteristics of mother-daughter relationships among Japanese female university students and their associations with students’ sexual risk-coping consciousness. We conducted a cross-sectional study using anonymous self-administered questionnaires. Participants were 329 female university students in their junior and senior years. The mother-daughter relationships of the study respondents were divided into the following three clusters: controlled group, close group, and independent group. The close and independent groups often consulted their mothers about sexual matters and, also, tended to be highly capable of actively expressing their opinions and cooperating with their partner in a sexual setting. In contrast, the controlled group revealed a significantly lower percentages of consulting their mothers on sexual matters and scored lower sexual risk-coping consciousness subscale scores. The controlled group may suffer a negative impact of the relationship with their controlling mothers as they try to build a good, trusting relationship with others. The characteristics of mother-daughter relationships may be associated with daughters’ sexual risk-coping consciousness.

## 1. Introduction

The average age when Japanese women, presently in their early 20s, first experience sexual intercourse was reported to be 18.6 years [1]. Approximately 37% of Japanese female university students have had sexual intercourse experience [2]. According to a survey of men and women between the ages of 20 and 59 years by one company, the LGBT (Lesbian, Gay, Bisexual, Transgender) community in Japan is reported to comprise 7.6% of the population [3], but there are no official statistics for the LGBTQ (Lesbian, Gay, Bisexual, Transgender, Questioning) community yet. Although not many sources provide details of the subject’s sexual orientation, it is presumed that the majority of Japanese women are heterosexual. In order to understand the current sexual behaviors of Japanese people, Sagami Rubber Industries, Co., Ltd. (Japanese condom manufacturer) conducted an internet survey targeting approximately 14,000 people in their 20s and 60s who were evenly distributed by gender and age [1]. As a result, about 15% of women in their 20s experienced their first sexual intercourse with a man they met via the Internet, such as a SNS (social networking service), social games, and dating sites. Additionally, about 16% of them experienced their first sexual intercourse with men who they were not dating [1]. Compared to the same survey conducted in 2013 [4], the numbers have increased, and the results suggested that the opportunity for young women to meet men and for being more accepting towards casual sex is increasing. In fact, according to a survey of the Japanese Ministry of Health, Labor and Welfare, the number of STIs (sexually transmitted infections) among women in their early 20s is higher than that of women of other ages and is on the rise [5]. In particular, the increasing of the number of chlamydia and syphilis patients is a serious issue. In addition, the number of abortions has been declining for women of other ages but has remained unchanged for women in their early 20s [6]. Therefore, for heterosexual Japanese women in their early 20s, it is important to maintain good sexual health, develop intimate relationships with a partner, and go through the process of having a family.

Among the numerous and varied relationships that individual experiences, there is little doubt that the mother-child relationship is the most influential for children’s social skills and behaviors [7]. Regarding an international comparative study, it was pointed out that young Japanese women tend to have a stronger emotional connection with their mothers than young American women [8]. In addition, the mother-daughter relationship that Japanese female university students perceive and experience has revealed that they have a high level of trust and intimacy [9,10]. Many healthcare behaviors are transmitted from parents to children and are learned as vital components of a healthy lifestyle. Kawata’s study [11] implied that the level of agreement between young women and their mothers on the method of obtaining axillary temperature measurements is significantly associated with presently secure mother-child relationships. Several studies also have indicated that health behaviors, including children’s prosocial and problematic behaviors, are influenced by relationships with caregivers; the associations between antismoking actions [12], treating caries [13,14], eating behaviors to avoid childhood obesity [15], and family relationships are said to be protective factors against unwanted teenage pregnancies [16,17]. Moreover, several studies have suggested that maternal education and practices are possibly effective in enhancing children’s health status [18,19].

It is also inferred that mother-daughter relationships influence daughters’ sexual behaviors. Previous studies have reported that mothers are a protective factor against early sexual initiation for girls [20] and that women in early adulthood with a good relationship with their mothers have a low likelihood of contracting STIs [21]. Studies have also significantly correlated conversations between mothers and daughters about sexual matters with the use of condoms and contraceptives [22] and have shown that these daughters are likely to talk to their physicians or other healthcare professionals about sexual problems [23]. In Japan, a study on the relationship between high school girls’ sexual intercourse experiences and the family environment reported that having a single parent, having a private room, and/or does not feel cozy when they are with one’s families were associated with increasing sexual intercourse [2]. In addition, it was reported that the lower the self-esteem as a family member, the higher the risk of contact with sexual information on the internet [24].

Having sex only when neither person nor the partner has a STI or having sex with only one partner is believed to be safe. Having sexual intercourse with a stranger or an unspecified number of people increases the risk of getting STIs or unintended pregnancy [25]. Based on these facts, it is necessary to highlight the importance of preventing STIs and unintended pregnancies among the younger generation and help them go through the process of developing intimate relationships with their partners while ensuring good sexual health. Kusano [26] defined the sexual risk-coping consciousness as recognition of the self-management ability to take appropriate actions to avoid risks in sexual relations and confidence in the sexual interpersonal ability to communicate intimately with others. The sexual risk mentioned here is a STI or an unintended pregnancy, and it was proved that those who have experienced sexual intercourse are more conscious of coping with sexual risk. This result suggested the need to consider appropriate sex education from the perspective of the development of interpersonal relationships. However, studies on the association between mother-daughter relationships and daughters’ sexual risk-coping consciousness are insufficient.

Therefore, this study aimed to understand the current situation of mother-daughter relationships among heterosexual Japanese women in their 20s, who many of become sexually active, as mentioned above, and to examine the association of these relationships with daughters’ sexual risk-coping consciousness and sexual behaviors. The definition of risky sexual behavior is as follows; having sexual intercourse with a stranger, an unspecified number of people, and/or without contraception. The study was designed to address two primary research questions: (1) What is the current situation of the Japanese mother-daughter relationship? and (2) Is a reliable relationship between a mother and a daughter associated with the daughter’s higher awareness of coping with sexual risk?

## 2. Materials and Methods

### 2.1. Study Design

This was an observational, cross-sectional study using an anonymous self-administered questionnaire. Participants were recruited using the snowball sampling method from four prefectures in Japan.

### 2.2. Participants and Procedures

The survey included female university students in their junior or senior years. A questionnaire was distributed between June and July 2019, and responses were collected until August.

The researchers discussed the date/time for distributing the questionnaire and the method/place for collecting responses with 4 universities and 12 university-affiliated clubs that agreed to participate in the study and visited them according to the set schedule. The purpose and method of this survey, protection of personal information, and freedom to withdraw were explained both in writing and verbally to the female students targeted by the survey. The self-administered questionnaire form and a collection envelope were distributed to those who provided their consent to participate in this study. The completed questionnaire was placed in the collection envelope and returned via a collection box or by mail.

In total, 636 questionnaire forms were distributed and 415 collected (response rate: 65.3%). After excluding those with missing data, 329 respondents were included in our analysis (effective response rate: 79.3%).

### 2.3. Contents of Questionnaire

The questionnaire comprised items pertaining to basic attributes, the mother-daughter relationship scale, sexual risk-coping consciousness scale, and actual sexual behaviors.

(1)Basic attributes

The basic attributes were age, marital status, department enrolled in, family structure, and family living together.

(2)Mother-daughter relationship scale [27]

This scale was used to measure the relationship between mother and daughter in late adolescence to early adulthood. It is composed of 35 items (6 items are reversed items) of 5 factors: “support for mother” (5 items, including “I want to take care of her”), “past conflicts” (6 items, including “We used to argue constantly”), “mother’s domination” (9 items, including “She forces her opinion on me”), “trust in mother” (10 items, including “I want to understand her feelings”), and “dependence on mother” (5 items, including “I tend to rely on her for everything”). Answers were rated on a five-point Likert scale from “very applicable” to “not applicable at all”.

The 5-factor scale included 35 items responded to on a 5-point Likert scale. The reliability coefficient (Cronbach’s α) for each of the original scales ranged between 0.78 and 0.90, indicating that the internal consistency of the scale was sufficiently high. The higher the score, the stronger the characteristic of the mother-daughter relationship associated with the factor. Permission to use the scale was obtained from the codevelopers.

(3)Sexual risk-coping consciousness [26]

This scale was used to measure respondents’ recognition of their self-management ability to take appropriate actions to avoid sexual risk. This 1-factor scale comprised 18 items responded to on a 4-point Likert scale. Cronbach’s α coefficient was 0.87, indicating high reliability. The higher the mean score, the higher the respondent’s sexual risk-coping consciousness. Permission to use the scale was obtained from the authors.

(4)Actual sexual behaviors

These questions pertained to respondents’ experience with sexual intercourse, the presence/absence of a sexual partner, frequency of contraceptive use, contraceptive method used, history of induced abortion, and history of STIs.

### 2.4. Statistical Analysis

For the mother-daughter relationship scale and sexual risk-coping consciousness, a factor analysis was performed to confirm the factor structure. Thereafter, scores on each factor were converted to standardized scores, and a cluster analysis was performed. The number of clusters was determined by examining the characteristics of the factor scores within each cluster and differences between the clusters. In addition, a Kruskal-Wallis test and multiple comparison test (Bonferroni correction) were performed to examine differences in the factor scores for each cluster. Each cluster name was discussed and determined by multiple researchers.

Fisher’s exact test and multiple comparison test (Bonferroni correction) were performed to examine differences in sexual behaviors between the mother-daughter relationship clusters, and a Kruskal-Wallis test and multiple comparison test (Bonferroni correction) were conducted to examine the scores for sexual risk-coping consciousness. A significance level of 5% (two-sided test) was used in all analyses. Software HAD 16.0 (https://norimune.net/had. Hiroshi Shimizu, Kansai Gakuin University, Japan, 2016) [28] was used for the cluster analysis, statistical software R [29] for the multiple comparison tests, and IBM SPSS ver. 25 for all other analyses.

### 2.5. Ethical Considerations

There are no official Japanese statistics on the LGBTQ community. The subject of this study is heterosexual women; however, we may have inadvertently asked members of LGBTQ community to participate in the study. Therefore, we explained that participation in this survey was completely voluntary, and it was not necessary to answer questions that they did not want to answer and added “I don’t want to answer” options to some questions.

Since the questionnaire included questions on aspects considered private, the completed form was collected in a sealed collection envelope.

This study was conducted with the permission of the Clinical Research Ethics Review Board of the university that the researchers are affiliated with. 

## 3. Results

### 3.1. Characteristics of Participants

Respondents’ ages ranged from 20 to 27 years, with a mean age of 20.9 (±0.93) years. The number of respondents enrolled in medical departments (medicine, dentistry, pharmaceutical sciences, nursing, etc.) was the highest at 224 (68.1%). The mothers of all the respondents were alive and well, and 43% of the respondents lived alone. As for sexual behaviors, 175 (53.2%) answered “they had experience of sexual intercourse”, and 131 (39.8%) answered that “they had a sexual partner”. Among the respondents who “had experience of sexual intercourse”, 80.3% said they used contraception “every time” and “almost every time”, 0.6% had a history of induced abortion, and 6.3% had a history of STIs (of which 80% had chlamydia and 20% genital candidiasis). None answered that they had been infected with condyloma acuminatum, gonorrhea, or syphilis (Table 1).

### 3.2. The Mother-Daughter Relationship Scale and Classification of Mother-Daughter Relationships

A factor analysis (principal factor method and promax rotation) was performed to confirm the factor structure of the mother-daughter relationship scale. In this study, 24 items were adopted (five items with a ceiling effect, five with a floor effect, and one with a factor loading less than 0.35 were deleted) and four factors (“positive feelings toward mother”, “control by mother”, “past conflicts”, and “dependency on mother”) extracted. Cronbach’s α coefficient for each of the four factors were 0.88, 0.84, 0.91, and 0.77, respectively (Table 2).

A cluster analysis (word method) was performed using the subscale scores of the scale to extract optimal clusters, and a Kruskal-Wallis test and multiple comparison test (Bonferroni correction) were conducted to examine differences in the factor scores for each cluster. As a result, the following three groups were identified: “controlled group”, “close group”, and “independent group” (Figure 1).

In Cluster 1, “control by mother” and “past conflicts” scores were high, and “dependency on mother” and “positive feelings toward mother” scores were low. Compared to others, this cluster indicated the highest “control by mother” and the lowest “positive feelings toward mother”; these participants formed the controlled group. In Cluster 2, since “positive feelings toward mother” and “dependency on mother” scores were high, and “control by mother” and “past conflicts” scores were low, it was named the close group. In Cluster 3, “past conflicts”, “control by mother”, and “positive feelings toward mother” scores were high, and the “dependency on mother” score was almost average. Since the “past conflicts” score of this cluster was the highest, the “control by mother” score was relatively high and “positive feelings toward mother” was slightly higher than others, while the “dependency on mother” score was lower than that of the participants of the close group; therefore, it was named the independent group.

### 3.3. Sexual Risk-Coping Consciousness

For sexual risk-coping consciousness, a three-factor structure was identified in this study consisting of “cooperative management type”, “active management type”, and “self-management awareness”. Cronbach’s α coefficient for each of the three factors was 0.82, 0.75, and 0.69, respectively (Table 3). 

### 3.4. The Degree of Respondents’ Consultation with Their Mothers about Sexual Matters in Each Mother-Daughter Relationship Cluster

The proportion of respondents who consulted their mothers about sexual matters was significantly lower in the “controlled group” than in the “close group” and “independent group” (close group: controlled group *p* < 0.001 and independent group: controlled group *p* < 0.05) (Figure 2).

### 3.5. Scores for Sexual Risk-Coping Consciousness in each Mother-Daughter Relationship Cluster

Among the mother-daughter relationship clusters, significant differences were noted in two factors of sexual risk-coping consciousness (*p* < 0.05 and *p* < 0.01); therefore, a multiple comparison test (Bonferroni correction) was performed. As a result, the score for the “cooperative management type” was significantly higher in the “close group” than in the “controlled group”, while, in the “active management type”, the score was significantly lower in the “controlled group” than in the “close group” and “independent group” (close group:controlled group *p* < 0.01 and independent group:controlled group *p* < 0.05) There were no differences between the clusters in “self-management awareness” in sexual situations (Figure 3).

### 3.6. Sexual Behaviors by Mother-Daughter Relationship Cluster

No significant difference was found between the clusters for “contraceptive use”, “history of induced abortion”, and “history of STIs”.

## 4. Discussion

### 4.1. Basic Attributes of the Study Participants

According to statistics on Japan, the number of female university students is highest in sociology departments, followed by medical departments and, then, by humanities departments. In this study, the highest proportion of students was from medical departments [30]. Regarding sexual behaviors, comparing statistics on Japan, the participants of this study were no different from average female university students in terms of their experiences with sexual intercourse and frequency of contraceptive use. However, a substantially lower percentage of our participants had a history of induced abortion or STIs [5,6].

In Japan, family environments such a having a single parent and having a private room were associated with increasing sexual intercourse [24]. In this study population, however, no respondents had a single parent. In addition, 43% of the respondents answered they lived alone, and the percentage was almost same as that of the previous report: 35.3–68.1% [31].

### 4.2. Validity of the Mother-Daughter Relationship Classification and Characteristics of Each Cluster

Among the mother-daughter relationship clusters, the “controlled group” had low levels of “positive feelings toward their mothers”, felt “controlled by their mothers”, and had experienced “past conflicts” with them. In addition, compared with the other groups, a significantly lower percentage of those in the “controlled group” consulted their mothers on sexual matters. A cluster analysis in a preceding study revealed that the group similar to the “controlled group” in the current study had little to no sense of being protected or accepted by their mothers, which, presumably, caused them to consider their mothers annoying or give up on them, leading to early separation [32]. In this study, it was inferred that the “controlled group” felt strongly controlled by their mothers, causing them to detach from them before they could reconstruct the relationship, which was affected by past conflicts.

In the “close group”, high scores were obtained for “positive feelings toward mother” and “dependency on mother”, while “control by mother” and “past conflicts” scored low. Furthermore, this group indicated “mother” as a consultant for sexual matters more than did those in the “controlled group”. 

Although the “independent group” had strong experiences with “past conflicts”, they scored significantly higher on “positive feelings toward mother” than did those in the “controlled group”. Furthermore, as shown in Figure 2, a higher percentage of respondents in this group indicated their mother as a consultant for sexual matters than those in the “controlled group”. As a characteristic of the relationship between a mother and young adult daughter, close contact and frequent negotiations can often cause them to hurt each other but can also nurture the strong sense of trust and intimacy between them, making it relatively easy to rebuild the relationship [9]. Additionally, the independent group in this study experienced conflicts with their mothers in the past, but currently had positive feelings toward them, indicating that their adversarial relationship was repaired. Furthermore, in the “controlled group”, which, like the “independent group”, scored high on “past conflicts”, their early separation from their mothers was associated with their low “dependency on mother”. However, in the “independent group”, the low “dependency on mother” was attributed to being mentally independent as a result of establishing a good relationship with their mother. Therefore, as a characteristic of the independent group in this study, it was inferred that, despite having experienced conflicts with their mothers in the past, their mother-daughter relationships were reconstructed in such a way as to nurture the daughters’ independence.

### 4.3. Associations of the Characteristics of Mother-Daughter Relationships with Daughters’ Sexual Risk-Coping Consciousness and Sexual Behaviors

The investigation of sexual risk-coping consciousness for each of the three mother-daughter relationship clusters revealed that the “close group” scored significantly higher for the “cooperative management type” of sexual risk-coping consciousness than the “controlled group”. Additionally, the “close group” and “independent group” scored significantly higher for the “active management type” than the “controlled group”.

“Cooperative management type” and “active management type” are subscales regarding self-confidence in terms of the sexual interpersonal relationship ability to form an intimate relationship and be able to communicate with a partner [26]. According to a preceding study, a higher-quality relationship between adolescents and their parents, especially between mothers and daughters, may help to protect against early sexual initiation. When female adolescents find their home environment enjoyable, frequent conversations with parents and a good family atmosphere prevent their early sexual initiation. [2]. Moreover, a higher percentage of women who find it difficult to communicate with their mothers regarding sexual intercourse and contraception choose a less effective contraceptive method than those who do not find this difficult [33]. Furthermore, it was also reported that, for adolescent girls, maternal–adolescent communication is the source of information on how to acquire condoms and contraceptives and their effectiveness [22]. 

In Japanese culture, it has been pointed out that young Japanese women tend to have stronger mother-child emotional ties than American women [8]. In addition, this study revealed that respondents in the close group who had reliable relationship with their mothers, and those in the independent group who rebuilt their relationships with their mothers, tended to consult and communicate with their mothers about sexual matters. As a result, they were also capable of building an equal relationship with their partners, which presumably has helped them avoid sexual risk by expressing their opinions in sexually high-risk situations. In contrast to these groups, the “controlled group” may suffer a negative impact of the relationship with their controlling mothers as they try to build a good, trusting relationship with others. These findings suggest that a good mother-daughter relationship is vital for fostering self-management abilities to avoid sexual risk among young women. To raise awareness about appropriate actions to avoid sexual risk, it is considered important to provide girls in early adolescence with support for maintaining a good mother-daughter relationship so that they can develop positive feelings toward their mothers. In addition, measures are needed to promote the reconstruction of the mother-daughter relationship represented by the “controlled group” in this study.

Our findings suggest the important role of mother-daughter relationships in maintaining and improving the sexual reproductive health/rights of women in young adulthood, which we consider is of significance in this study.

### 4.4. Limitations

Since this study did not use a random sampling method, it is difficult to generalize the results thereof. For sexually risky behaviors, no significant differences were found between the three mother-daughter relationship clusters. The reason we did not find a significant difference between contraception use, experience of abortion, and STIs and mother-daughter relationships is because it is presumed that there are several types of contraceptive methods and no specific answer was obtained in this survey and that the number of respondents who experienced abortions and STIs was very low.

In addition, because the study targeted young women who had a mother, the marital status of the mother was not questioned. However, the fact that the presence or absence of a father and the impact of mother-father relationships were not examined is considered a limitation of this study. Furthermore, previous studies reported that having a single parent, private room, working mother, older male sibling, and friend of the same age and gender are factors influencing the sexual behaviors of young adults [2,33]. As various factors are considered influential in the sexual behaviors of young people, further studies are necessary.

## 5. Conclusions

The following three clusters were identified regarding our respondents’ mother-daughter relationships: the controlled group, the close group, and the independent group. Regarding sexual risk-coping consciousness, the close group and the independent group had higher levels of awareness, which allowed them to avoid sexual risk by expressing their opinions when communicating with their partners, and this ability might be influenced by young Japanese women’s positive and healthy relationships with their mothers.

One of the main findings of this study is that sexual literacy and education toward self-management may develop close communication between the mother and daughter and infuse trust and intimacy in the relationships. Our findings suggest that, as a specialized profession, we need to impress on society the importance of a good mother-daughter relationship so that young women can develop self-management abilities to avoid sexual risks.

## Figures and Tables

**Figure 1 ijerph-17-08795-f001:**
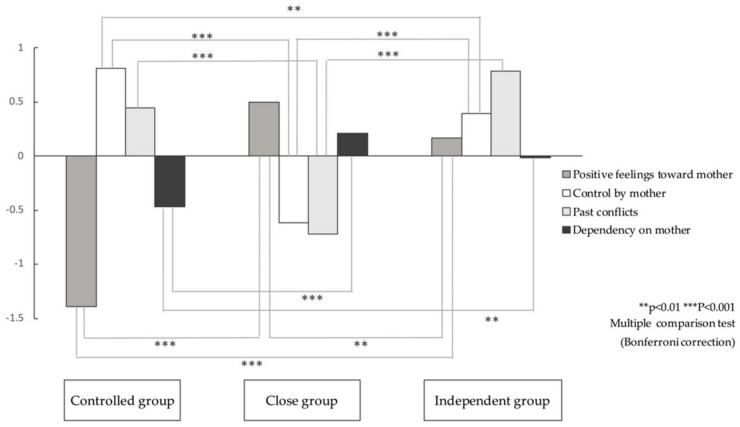
Mother-daughter relationship scale: cluster analysis results (*n* = 329).

**Figure 2 ijerph-17-08795-f002:**
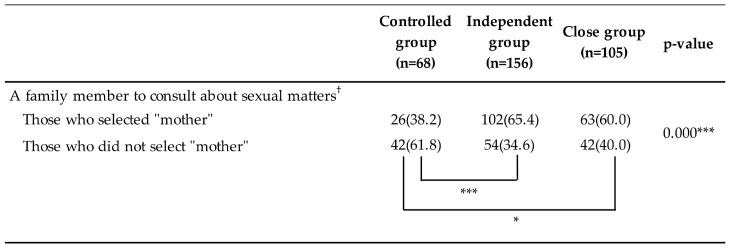
Mother-daughter relationship clusters and percentage of those who indicated their “mother as a consultant” in each cluster (*n* = 329). ^†^ Options: mother, father, brother, sister, and other. Fisher’s exact test and multiple comparison test (Bonferroni correction). The number is the number of respondents, and the value in ( ) is the %. *** *p* < 0.001, * *p*< 0.05.

**Figure 3 ijerph-17-08795-f003:**
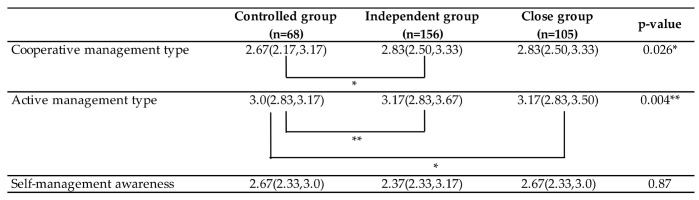
Examination of sexual risk-coping consciousness scores in each mother-daughter relationship cluster (*n* = 329). Kruskal-Wallis test and multiple comparison test (Bonferroni correction); The value is the median, and the values in ( ) are the 25th and 75th percentiles; ** *p* < 0.01 and * *p* < 0.05.

**Table 1 ijerph-17-08795-t001:** Basic attributes (*n* = 329).

Attributes	Options	Minimum	Maximum	Mean ± SD	*n* (%)
Age		20	27	20.9 ± 0.9	
Department	Medical				224 (68.1)
	Other				105 (31.9)
Family living together	Living alone				141 (43.0)
(multiple answers)	Living with family				176 (53.3)
	Other				12 (3.7)
Family structure	Father				286 (86.9)
(multiple answers)	Mother				329 (100)
Experience of sexual intercourse	Yes				175 (53.5)
	No				118 (36.1)
	No response				36 (10.9)
Age of first sexual intercourse encounter (*n* = 174 *)		12	21	18.6 ± 1.6	
Sexual partner currently dating	Yes				131 (39.9)
	No				181 (55.2)
	No response				17 (5.2)
Sexual partner currently not dating	Yes				33 (10.8)
	No				251 (82.3)
	No response				45 (13.7)
Number of sexual partners currently not dating (*n* = 32 ^†^)		1	4	1.8 ± 1.0	
Frequency of contraceptive use	Every time				119 (68.0)
(*n* = 175 ^‡^)	Almost every time				39 (22.3)
	Rarely				10 (5.7)
	Never				2 (1.1)
	Currently not engaged in sexual intercourse				4 (2.3)
	No response				1 (0.6)
History of induced abortion	Yes				1 (0.6)
(*n* = 175 ^‡^)	No				172 (98.3)
	Don’t want to answer				2 (1.1)
History of sexually transmitted infections (multiple answers)	Chlamydia				2 (1.1)
	Candida				8 (4.6)
	Others ^§^				0 (0)
(*n* = 175 ^‡^)	Never				164 (93.7)
	No response				7 (4.0)

* Those who answered they had experience with sexual intercourse (one was no response); ^†^ Those who answered they had a sexual partner they were currently not dating (one was no response); ^‡^ Those who answered that they had experience with sexual intercourse; ^§^ Syphilis, gonorrhea, condyloma acuminatum, genital herpes, and HIV.

**Table 2 ijerph-17-08795-t002:** Mother-daughter relationship scale: factor analysis results (*n* = 329).

Item	α
Factor 1: Positive feelings toward mother (10 items)	0.88
Factor 2: Control by mother (6 items)	0.84
Factor 3: Past conflicts (4 items)	0.91
Factor 4: Dependency on mother (4 items)	0.77

**Table 3 ijerph-17-08795-t003:** Sexual risk-coping consciousness: factor analysis results (*n* = 329).

Item	α
Factor 1: Cooperative management type (6 items)	0.82
Factor 2: Active management type (6 items)	0.75
Factor 3: Self-management awareness (6 items)	0.69

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
