# Peer review of "Characteristics of Mother–Daughter Relationships and Sexual Risk-Coping Consciousness among Japanese Female University Students"

_ijerph, 2020, doi:10.3390/ijerph17238795_

Round 1

Reviewer 1 Report

I would appreciate a more deep analysis of sexual behaviours and an explanation to not have found significant difference between the clusters regarding to this factor 

The discussion cites a previous study that links the difficulty of daugthers to communicate with their mothers with the less effective contraceptive method chosen. I woul like to see any implications for the current study that at the same time links the independent group with an queal relationshio with their partners. To what kind of choices do the last one refer to?

A cross cultural analysis would be interesting as France and Anglo cultures' studies -e.g. in Pennsylvania, Bahamas- are cited

One of the main findings for sexual literacy and education toward self-managament  is that close contact and negotiation between mother & daugther brings trust and intimacy

Reviewer 2 Report

Overall, the manuscript provides valuable cultural information regarding the variables mother-daughter relationships and the Sexual Risk-Coping Consciousness among Japanese young females. While the design and the statistical analyses are appropriate and clearly presented, the authors could improve the Introduction and the Discussion parts of the manuscript. In line with this, several suggestions are listed bellow:

Lines 11-12 - Abstract  and Lines 31-32 - Introduction - The statement "It is important for women in their 20s..." indicates that the authors do believe that only heterosexual relationships are a desirable norm for young women, as well as having a family. Would it be possible to reformulate this statement in a more permissible way to gender diversity?

Lines 35-36 - please provide references to the statement "On the other hand, having sexual intercourse with a large number of random people...."

Line 43 - please provide references to the statement "It is inferred that mother-daughter relationships influence daughter's sexual behaviors". There are several studies in the literature on the topic of mother-daughter relationship in relation to sexual health and sexual risk behaviors.

Introduction - I suggest that the authors should include in their introduction more key concepts than the ones that they have already presented, such as: parental influence process, intergenerational transmission of sexuality knowledge, parental attitudes towards sexual behaviors of their children, parent-child communication etc. 

Lines 228-230 - Please provide reference to the statement "According to a preceding study,..."

Lines 267-269 - The last statement of the Conclusions part could be more elaborated in terms of clearly presenting what a functional mother-daughter relationship means in relation to the findings of the study. 

Reviewer 3 Report

Dear Editor and Authors:

This study aims to check if there is a relationship between characteristics of mother-daughter relationships and sexual risk - coping consciousness among Japanese female university students. As strengths, raising a relationship between the mentioned variables is original. Likewise, the effort of the authors can be observed when carrying out this study. In general, authors should further deeper into the Introduction. They are based on little bibliography. In addition, in that section they should include the definition of the study variables. Further, the information must be orderly. The sample used is not too large and should be described in more detail. It should be mentioned that there is only one general objective, but the analyzes that are carried out are on specific occasions and do not follow the line of this general objective. Therefore, it is recommended to add specific objectives that support those analyzes and results. Likewise, the inclusion of hypotheses is recommended. In addition, these objectives and hypotheses can help to build the order of the Results and Discussion sections, and in this last section, know if these objectives have been confirmed or not.

Next, I make specific comments and recommendations:

Abstract
Page 1/11: "For women in their early 20s, it is important to go through the process of developing intimate relationships with men and having families while ensuring their health."

- This sentence is incomplete. These women should be described, for example, where they are from, sexual orientation, etc.

Introduction

Page 1/11: It is important for young people aged 20 years and older to find a partner to have a sexual relationship with in the process of building a good intimate relationship and having a family [1].

- Please add more data on the characteristics of these young people (example, sex, sexual orientation, place of residence, among others). This may be determining the importance given to having sex at that age and building a family. Furthermore, authors should rely on more literature to support this phrase.

Page 1/11: "Today’s widespread use of the Internet and smartphones has increased meeting opportunities for men and women, allowing them to be freely engaged in romantic relationships."
- Please cite this sentence. Do people physically (i.e. face-to-face) date more than before on social media and smartphones? Is it possible that people are isolating themselves by these devices and types of relationships through social media? Please clarify these issues in the text and support your explanations in the bibliography.

Page 1/11: "On the other hand, having sexual intercourse with a large number of random people is believed to increase the risk of sexually transmitted infections (STIs) and unwanted pregnancy."
- Please cite this sentence. In addition, the authors must give specific data. It is inappropriate to mention that "people is believed to increase ..."

Page 1/11: "In Japan, the number of cases of STIs is higher among women in their early 20s than those in other age groups, and it is on an increasing trend."
- Please cite this sentence.

Page 1/11: One of the variables under study are risky sexual behaviors. Therefore, the definition of risky sexual behavior should appear before mentioning its consequences (ie., unplanned pregnancies, sexually transmitted infections, etc.).

Page 1/11: "Based on these facts, it is necessary to highlight the importance of preventing STIs and unintended pregnancy in young women and help them go through the process of developing intimate relationships with men while ensuring their health."
- What about young men? This should be included in the text.

Page 2/11: "In Japan, however, studies on the association of mother-daughter relationships with daughters’ sexual health promotion are insufficient."
- How many investigations are there? Why is it insufficient? What are the authors based on to affirm this? Please clarify this in the text and include bibliographic citations.

Page 2/11: "...when many become sexually active..."
- This should be based on bibliography. Please cite this information.

Page2/11: What hypotheses are raised for this study?

Page 2/11: "Our findings suggest the important role of mother-daughter relationships in maintaining and improving the sexual reproductive health/rights of women in young adulthood, which we consider is the significance of this study."
- This paragraph should not be part of the introduction, as it precedes the results section. Therefore it is recommended to include it in the most appropriate place (e.g., discussion section).

Materials and Methods
Page 2/11: "This was an observational, cross-sectional study using an anonymous self-administered questionnaire".
- Please quote this information.

Page 2/11 - Line 78: Was the sexual orientation of the women in the present study evaluated? Throughout the text, it is assumed that women are going to have sex with men. Please clarify this aspect.

Page 2/11 - Line 78: Please indicate in the text what is evaluated with "family structure".

Page 2/11 - Line 78: "...family living together...". Do the authors refer to the number of people living in the same house? Please clarify this in the text.

Page 2/11 - Line 82: Include the name of the factors and an example item for each factor.

Page 2/11 - Lines 83-84: As it is written, it is not known to which factor each reliability belongs. Therefore, it is recommended to place the reliability in range, for example: "The reliability of the factors ranged between .78 and .90."

Page 2-3/11: Another question that should be clarified in the text is whether the reliabilities are from the original scale or from the sample of the present study.

Pages 3, Tables 2 and 3: The factor analyzes of the two scales are not necessary, presenting the reliability for the sample of the present study is sufficient.

Page 3/11 - Lines 117-119: This information should be taken to the procedure section.

Page3/11 - Lines 120-129 and Table 1: The characteristics of the participants must appear in the participants section of the "Materials and methods" section.

Page 5/11: Left justify the column "Item".

Page 5/11 - Lines 141-145: Please, for the better understanding of the results, it is recommended to write a description of each group.

Discussion
The discussion should be carried out according to objectives and hypotheses. The inclusion of hypotheses has already been mentioned previously. It is not entirely clear what this study contributes. This is what the authors should focus on, giving more importance to their data.

Reviewer 4 Report

There is no specific comments for authors.

Author Response

We appreciate the time and effort you and each of the reviewers have dedicated to providing insightful feedback on ways to strengthen our paper.

Round 2

Reviewer 2 Report

The authors have carefully addressed all the reviewer's suggestions. References were added in the body of the text according to the first review report. I do consider that the manuscript is now acceptable for publication.

Author Response

Thank you very much for providing important comments. We are thankful for the time and energy you expend.

Reviewer 3 Report

Dear Editor and Authors:

The manuscript has been improved. I have been able to observe the efforts of the authors to carry out this manuscript and the recommendations indicated by the reviewers. There are some minor issues that need to be fixed:

Introduction

Page 1 – Line 40: “by one 39 Japanese company…”. What company? It would be interesting if the company name appeared at least in parentheses.

Page 1 – Line 40: “about 15% of women in their 20s”. How many women were evaluated? In this case, data on these women are also lacking. Please include a brief description of these women.

Page 2 – Lines 44-45: “and it was considered that the opportunity for young women to meet men and more free love is possible, and is increasing.” This explanation is not adequate. In addition, it should be supported by bibliography. Please fix this issue. Another question that can be mentioned in this sentence is the following: What do the authors mean by "more free love"? Love should be free in all its forms. Please clarify this.

Page 2: The definition of risky sexual behaviors still does not appear.

Discussion:

Page 11 – Line 302: “…help suppress their sexual behaviours”. Please reconsider this explanation. Since perhaps these young women are not repressing their sexual activity. Perhaps what happens is that by normalizing sexuality in their family and being able to talk about it openly they learn more preventive methods. For this reason, the sexual behaviors of these young women are safer and the consequences of risky behaviors are reduced.
